# Lower Respiratory Tract Infections in Pediatric Patients with Severe Neurological Impairments: Clinical Observations and Perspectives in a Palliative Care Unit

**DOI:** 10.3390/children9060852

**Published:** 2022-06-08

**Authors:** Maximilian David Mauritz, Carola Hasan, Pia Schmidt, Arne Simon, Markus Knuf, Boris Zernikow

**Affiliations:** 1Paediatric Palliative Care Centre, Children’s and Adolescents’ Hospital, Witten/Herdecke University, 58448 Witten, Germany; c.hasan@kinderklinik-datteln.de (C.H.); p.schmidt@kinderklinik-datteln.de (P.S.); b.zernikow@kinderklinik-datteln.de (B.Z.); 2Department of Children’s Pain Therapy and Paediatric Palliative Care, Faculty of Health, School of Medicine, Witten/Herdecke University, 58448 Witten, Germany; 3Pediatric Oncology and Hematology, University Hospital Homburg Saar, 66421 Homburg, Germany; arne.simon@uks.eu; 4Department for Pediatric and Adolescent Medicine, Worms Clinic, 67550 Worms, Germany; markus.knuf@klinikum-worms.de; 5Pediatric Infectious Diseases, University Medicine, 55131 Mainz, Germany

**Keywords:** lower respiratory tract infection, pneumonia, severe neurological impairment, children with medical complexity, children, adolescents, pediatric palliative care

## Abstract

Pediatric palliative care (PPC) patients with a severe neurologic impairment (SNI) suffer considerable morbidity and increased mortality from lower respiratory tract infections (LRTIs). The indication and choice of antibiotic therapy for bacterial LRTIs are often challenging given the lack of evidence-based treatment recommendations for this vulnerable patient population. We conducted an observational study before the SARS-CoV-2 pandemic in an eight-bed pediatric palliative care inpatient unit. During two years of surveillance, we diagnosed and treated 33 cases of a bacterial LRTI in patients with an SNI; 5 patients were hospitalized with an LRTI more than once. Two patients died from complications due to LRTIs during hospitalization. Three patients (15%) were colonized with multidrug-resistant organisms. An initial antibiotic treatment failed in one-third of the cases; a successful therapy of the LRTI was achieved with broad-spectrum and extended-spectrum penicillins (*n* = 13; in combination with β-lactamase inhibitors for *n* = 5 cases), cephalosporins (*n* = 13: *n* = 4 second-generation and *n* = 9 third-generation cephalosporins; in combination with other substances for *n* = 5 cases), ciprofloxacin (*n* = 3), and meropenem plus vancomycin (*n* = 2) or meropenem (*n* = 1). A respiratory specimen was obtained in 66.7% of cases with *P. aeruginosa*, *E. coli*, and *K. pneumoniae* accounting for the majority of the detected species. In most cases, there was no definite confirmation that the LRTI was caused by the species detected. The diagnostics and treatment of bacterial LRTIs in PPC patients with an SNI are challenging. The lack of controlled studies and the heterogeneity of this population often necessitate an individual approach. This lack of controlled studies may partly be compensated by a set of diagnostic and antibiotic stewardship criteria.

## 1. Introduction

Pediatric palliative care (PPC) focuses on severely or terminally ill children and adolescents as well as their families. It aims to provide medical, nursing, psychological and social support in accordance with the individual situation of the patient. Although the majority of these patients can be described as “children with medical complexity” (CMC) [1,2], a considerable proportion has an irreversible progressive or a congenital non-progressive neurological impairment [3,4,5]. Characteristic clinical examples are severe cerebral palsy from a hypoxic or an ischemic brain injury, congenital encephalopathies, neuromuscular diseases, or progressive metabolic disorders [6,7]. Most of these patients experience a severe neurological impairment (SNI), defined by the combination of a motor impairment, medical complexity, and the need for assistance with the activities of daily life resulting from the aforementioned complex chronic conditions [8].

This group of patients is particularly susceptible to infections, especially lower respiratory tract infections (LRTIs) [9,10]. The causes of the high rate of LRTIs are complex, multifactorial, and differ from patient to patient in this heterogeneous group of children and adolescents. Depending on the age and pre-existing immunity, most LRTIs are primarily caused by respiratory viral pathogens; the respiratory syncytial virus (RSV), the influenza virus, and SARS-CoV-2 may cause severe LRTIs, particularly in children with an SNI [11,12,13]. In addition to these viral airway infections, this group of patients is also particularly prone to bacterial LRTIs. In clinical practice, it remains a challenge to distinguish viral from bacterial LRTIs in pediatric patients with an SNI because both may cause the same clinical signs of pneumonia [14]. 

Individually, many patients with an SNI are impaired by a reduced airway clearance; this may be due to impaired coordination and positioning, a restrictive ventilation disorder (e.g., due to scoliosis), or respiratory muscle weakness. These patients are particularly vulnerable to develop atelectasis due to the retention of bronchial secretions. An atelectasis developed in this way yields a higher risk of a bacterial superinfection for these patients [15,16].

In addition, micro- and macroaspiration due to dysphagia, oromotor dysfunction, seizures, and gastroesophageal reflux may play a role [17,18,19,20,21]. Gastric acid suppression, in particular by proton pump inhibitors, may foster the oropharyngeal and gastric colonization with bacterial pathogens [22]. Patients whose SNI is associated with prematurity also tend to have more frequent and severe lower respiratory tract infections due to chronic lung disease of prematurity [23]. A subset of pediatric patients with SNI also underwent tracheostomy, associated with an increased risk of tracheo-pulmonary infections. The tracheostomy tube bypasses the naturally protective nasal and oral airway and is inevitably colonized with biofilm-producing bacteria [24]. Additionally, difficulties concerning the timely recognition of clinical LRTI signs may cause delayed initiation of adequate therapy [25]. 

The aforementioned aspects lead to a higher susceptibility to bacterial LRTIs like community-acquired pneumonia and aspiration pneumonia; directly from the oropharynx and indirectly from gastric contents [26]. Hereby, otherwise harmless colonization can exacerbate and lead to an infection. Considering the medical history of CMC, close contact with specialized in- and outpatient nursing care, dependence on medical devices, and multiple cycles of broad-spectrum antibiotic treatment eventually yield a higher share of colonization and subsequent infection with multidrug-resistant organisms (MDRO) [27]. Misinterpretation of colonizing bacteria and their in vitro sensitivity may lead to overtreatment with broad-spectrum, second or third-line antibiotics. 

In sum, these factors result in bacterial LRTI as a cause of considerable morbidity and a major cause of mortality [28,29,30], significant distress from dyspnea [31], and often progressive respiratory failure [32] secondary to severe recurrent LRTI in this group of patients. For treatment, PPC patients with SNI and respiratory infections often require hospitalization [33,34,35,36]. Inpatient treatment is usually provided in a general pediatric or pediatric neurology ward. In our report, we describe patients treated in our specialized PPC unit.

Despite the high incidence of LRTI in this patient population, patients with SNI are not included in current guidelines for the treatment of pneumonia in children and adolescents, which refer to otherwise healthy children without CMCs [37,38]. Since no evidence-based treatment recommendations exist for the antibiotic treatment of this group of patients, LRTI management presents significant challenges: treatment according to standard regimens and adherence to antibiotic stewardship (ABS) versus consideration of the increased incidence of colonization with Gram-negative bacteria and MDRO [27] with early escalation of antibiotic therapy. This study aimed to investigate the microbial diagnostics, antibiotic therapy, and outcome of inpatient treatment for LRTI patients with SNI in PPC.

## 2. Materials and Methods

### 2.1. Design and Setting

The PPC unit at the Children’s and Adolescents’ Hospital, Datteln, Witten/Herdecke University, Germany, is a self-contained palliative care unit in a tertiary care children’s hospital. It consists of eight single rooms and provides intensive pediatric palliative hospital care. Annually, approximately *n* = 200 children are admitted to the unit. Three-quarters of patients have an underlying neurological disease [4].

### 2.2. Patients and Definition of Bacterial LRTIs

Between 1 February 2018 and 31 January 2020, all patients were screened for MDRO. All patients with an SNI, as described in the Delphi consensus-based definition [8], and treated for LRTIs were included in this study. LRTIs were diagnosed by an attending physician based on the clinical signs (tachypnea, fever, coughing, dyspnea, increased respiratory secretions, new-onset or an increased demand for supplementary oxygen, or pneumonic rales on auscultation), lab tests, and—if available—radiological criteria [24]. 

### 2.3. Data and Material Collection

All patients were screened for MDRO upon admission. Swabs were taken from the nostrils, throat, rectal/perirectal region, and—if applicable—the entrance sites of devices (e.g., gastrostomy). We used a smear utensil set (Nerbe Inc., Winsen/Luhe, Germany), including a flocked plastic swab and an Amies Agar Gel medium without charcoal for the MDRO screening. The respiratory secretions (including tracheal secretions in patients with a tracheostomy) were sampled with a tracheal suction kit (Dahlhausen Inc., Cologne, Germany). From September to March, patients with new-onset respiratory symptoms were also screened by polymerase chain reaction (PCR) tests from throat swabs for the influenza virus, RSV, parainfluenza virus, and human metapneumovirus. Additionally, medical records were reviewed for the treatment course of each patient.

### 2.4. Antibiotic Therapy

Inpatient antibiotic therapy was prescribed by the on-duty physician in the emergency department or the attending physicians in the PPC unit. Outpatient oral antibiotic therapy was prescribed by the attending office-based pediatrician. The conduct of this study did not affect the choice or duration of antibiotic therapy.

### 2.5. Statistical Analyses

The descriptive statistics were obtained using IBM SPSS Statistics (Version 28.0. Armonk, NY, USA).

## 3. Results

### 3.1. Patient Characteristics

During the 2-year period, 20 PPC patients with an SNI were treated as inpatients due to a bacterial LRTI in our pediatric palliative care unit on 33 occasions. Five patients were hospitalized with a bacterial LRTI more than once during the period (range: 2–5 admissions). The median age was 6.2 years (range: 0.4–21.5 years) and 12 patients (60%) were male. The average length of stay in the hospital was 14.5 days. A chest radiograph (CXR) was performed on 26 occasions (78.8%); radiological findings indicating an LRTI were present in all cases. As possible risk factors, 18 patients (90%) had epilepsy (mainly structural or symptomatic focal epilepsy) and 12 patients (60%) had cerebral palsy, all of whom were classified by the Gross Motor Function Classification System (GMFCS) as level V [39]. All patients received enteral feeding via a gastrostomy tube. One patient had a tracheostomy, through which they received nocturnal assisted ventilation due to myopathy. Another patient received a nasal high-flow therapy due to an upper airway obstruction and chronic respiratory failure. Detailed patient characteristics are shown in Table 1. 

Four patients (20%) were colonized in their gastrointestinal tract (rectal swabs) with MDRO (MDR *Escherichia coli* (*n* = 2) and MDR *Morganella morganii* (*n* = 1), both with resistance against piperacillin, third-/fourth-generation cephalosporins, and fluoroquinolones). Methicillin-resistant *Staphylococcus aureus* (MRSA) was detected in rectal swabs and a respiratory specimen in one patient.

Treatment in the pediatric intensive care unit (PICU) was required for six patients. Two patients died as inpatients: a male 9.6-year-old patient with hypoxic–ischemic encephalopathy following perinatal asphyxia died of respiratory failure in the PICU due to the palliative situation; the non-invasive ventilation started in the PICU was not escalated to invasive ventilation. Another male 2.9-year-old patient with holoprosencephaly died of severe sepsis after the completion of an initially successful antibiotic treatment for a bacterial LRTI. Appendix A shows the detailed information on the underlying diseases and secondary diagnoses.

### 3.2. Isolated Pathogens from Respiratory Specimens

Respiratory material was obtained from 22 (66.7%) cases, mainly sputum and tracheal secretions. Seven (31.8%) respiratory cultures showed no bacterial growth. The remaining 15 samples contained 23 different bacterial isolates. *Pseudomonas aeruginosa*, *Escherichia coli*, and *Klebsiella pneumoniae* were the most prevalent. Details on the isolated bacteria from the respiratory specimens are shown in Table 2. Three patients harbored a respiratory virus (parainfluenza: *n* = 2; RSV: *n* = 1).

### 3.3. Antimicrobial Therapy

#### 3.3.1. Outpatient Antibiotic Therapy

Eight (24.2%) patients had already received an outpatient oral antibiotic therapy before admission to the PPC unit consisting of cephalosporins (cefixime, cefuroxime, or cefaclor, the latter in combination with erythromycin), amoxicillin, amoxicillin–clavulanic acid, or ciprofloxacin. At admission, several of the outpatient therapies were changed. In the case of ciprofloxacin and cefuroxime, these therapies were continued. Therapies with amoxicillin and cefaclor were changed to ciprofloxacin. The amoxicillin–clavulanic acid therapy was changed to ampicillin–sulbactam intravenously administered; therapies with cefixime were switched to ceftazidime plus ampicillin–sulbactam plus clarithromycin and cefaclor was changed to ceftazidime plus gentamicin.

#### 3.3.2. Initial and Subsequent Inpatient Antibiotic Therapy

In 24 (72.7%) cases, the initial inpatient antibiotic therapy was intravenous. The oral therapy (*n* = 9; 27.8%) consisted of amoxicillin, ciprofloxacin, or clarithromycin. The intravenous therapy consisted of ampicillin; in several instances, with clarithromycin or combined with sulbactam and cephalosporins. Details on the antimicrobial therapy can be found in Table 3. An initial therapy with cefuroxime (*n* = 4), clarithromycin (*n* = 2), ampicillin, ampicillin plus ciprofloxacin, cefuroxime plus erythromycin, cefotaxime, and ceftazidime plus gentamicin (each *n* = 1) failed in one-third of cases. In two cases, the initial and the subsequent therapy were changed. In these cases, the initial therapy of cefuroxime was switched to ciprofloxacin and treatment with ampicillin was changed to ceftazidime plus gentamicin. These therapies were subsequently switched to a successful therapy with piperacillin–tazobactam and meropenem plus vancomycin, respectively. Eventually, the therapy was escalated to broad-spectrum antibiotics in a number of patients for a successful treatment. An oral therapy was successful in 10 cases and consisted of amoxicillin, cefpodoxime, or ciprofloxacin. The intravenous therapy (*n* = 22) consisted of broad-spectrum and extended-spectrum penicillins (partly in combination with β-lactamase inhibitors), cephalosporins (in two cases, in combination with clarithromycin), and meropenem or meropenem plus vancomycin. In two of three cases, meropenem was used to treat patients colonized with multidrug-resistant Gram-negative bacteria (MDR-GNB). Vancomycin was empirically used on two occasions in combination with meropenem after treatment attempts with ceftazidime and an aminoglycoside failed. There was no evidence of methicillin-resistant pathogens in these cases. 

#### 3.3.3. Successful Inpatient Antibiotic Therapy

The successful therapy of LRTIs was achieved with broad-spectrum and extended-spectrum penicillins (*n* = 13; in *n* = 5 cases, in combination with β-lactamase inhibitors), cephalosporins (*n* = 13: *n* = 4 second-generation and *n* = 9 third-generation; *n* = 5 in combination with other substances), ciprofloxacin (*n* = 3), and meropenem plus vancomycin (*n* = 2) or meropenem (*n* = 1). The sequence of antibiotic medications and successful therapies are displayed in Figure 1.

#### 3.3.4. Comparison of In Vitro Sensitivity and Successful Therapies

We compared the in vitro sensitivity of the detected pathogens in the respiratory specimens with the effective antibiotic therapies in 15 cases. In ten (66.7%) instances, the bacteria were sensitive to the antibiotics used. In five (33.3%) cases, the pathogens detected in the respiratory specimens were resistant to the antibiotic therapy that successfully treated the LRTI. These were (pathogen detected–antibiotic used) *Escherichia coli*–cefuroxime, *Serratia marcescens*–cefpodoxime, *Acinetobacter baumannii*–cefuroxime, *Klebsiella pneumoniae*–cefuroxime, and MRSA–ampicillin, respectively.

## 4. Discussion

During the two-year observation period, we treated 33 cases of bacterial LRTI in 20 PPC patients with SNI. Overall, 165 individual patients were admitted on 386 occasions [40]. Inpatient treatment of bacterial LRTI accounted for 8.6% of all admissions or 12.1% of all patients, respectively. This observation is consistent with the high proportion of morbidity associated with pulmonary infections in pediatric patients with SNI reported in the literature [28,29,30]. PPC patients have a distinctive higher incidence of colonization with MDRO [27]. Likewise, 20% of our patients were colonized with MDR-GNB or MRSA. 

A range of antibiotic therapies was used. In one-third of cases, the initial antibiotic therapy failed; for two of these cases, the treatment had to be changed twice. The wide range of initial therapies and escalation strategies (in cases of initial therapy failure) was most likely due to the lack of guidelines for treating LRTIs in children with CMC and SNIs [41]. The severity of the bacterial LRTI in these patients may also have influenced the choice of individual treatment [42]. A large proportion of patients were successfully treated with aminopenicillins in accordance with the guidelines for otherwise healthy children [37,38]. 

### 4.1. Oral versus Parenteral Antibiotic Therapy

A large proportion received an oral antibiotic therapy. This practice relates to the difficulties with venous access in this group of children as well as the availability of a nasogastric or a gastrostomy tube [43]. One patient had a port-a-cath catheter implanted due to these difficulties and required repeated intravenous antibiotic therapies during the observation period. There have been promising results demonstrated for successful oral antibiotic therapies for severe pneumonia [44], even for infections with MDR-GNB [45].

On the other hand, infections such as pneumonia caused by MDRO or *Pseudomonas aeruginosa* may not be treatable with oral antibiotics. Here, fluoroquinolones such as ciprofloxacin often remain the only (off-label) oral therapy option [46,47]. Most side effects of fluoroquinolones in children are not persistent and, according to recent studies, a few may occur just as frequently as other antimicrobial agents [48,49]. However, the use of fluoroquinolones is also associated with an increased rate of MDRO colonization and/or infection in hospitalized children [50]. Their use should be limited to cases where other substances are ineffective or not tolerated. In these cases, special attention must be paid to the dosing and the occurrence of adverse effects [49]. 

### 4.2. Relevance of Respiratory Specimens

The microbiology results from the respiratory tract specimens were not always helpful in guiding the antibiotic therapy; in a few cases, the LRTI was successfully treated although the detected bacterial species were resistant to the antibiotics used. Obtaining a representative sample from the lower respiratory tract can be difficult in non-intubated patients and contamination with organisms that colonize the upper respiratory tract (or a tracheostomy) is common [51]. The collection of specimens from the lower respiratory tract—for example, by obtaining induced sputum—could improve the diagnostic value of the findings in pediatric patients with bacterial LRTIs, but may not be feasible in patients with an SNI [52]. Bronchoscopy with bronchoalveolar lavage may provide a representative sample from the lower respiratory tract, but is not always warranted or possible in the context of the palliative nature of the disease or its complexity.

### 4.3. Antibiotic Stewardship and Impact of MDRO on Antibiotic Therapy

The broad spectrum of pathogens, especially MDRO, poses an additional challenge to the therapy. An increase in the proportion of MDRO demands the development of new antibiotic substances [53]. In two cases, MDR-GNB colonization may have affected the escalation of the antibiotic therapy with carbapenems, even though MDROs were not detected in any respiratory specimen. The evolving proportion of MDRO in pediatric patients complicates antibiotic therapies in an increasing number of cases [54,55]. Attending physicians tend to “err on the side of caution” in patients colonized with MDR bacteria and primarily use broad-spectrum antibiotics. 

In addition to efforts to prevent colonization and infection with hospital-acquired MDROs [40,56], antibiotic therapies in this patient population clearly foster the selection of MDROs. Although most reports investigating these issues, including ABS in children with SNIs, derive from secondary (post-acute) or long-term care facilities [57,58,59], ABS programs can reduce the occurrence and spread of MDRO in the long term [50,60]. Avoiding unnecessary antibiotic therapies also plays a vital role in this context as these patients are also affected by severe viral respiratory infections [11,12,13]. Other causes should be actively excluded such as urinary tract infections (especially in the presence of a urinary catheter or an existing urine transport disorder), skin and soft tissue infections, or ear, nose, and throat infections, including sinusitis [57,61]. Each avoided MDRO colonization prevents limitations for PPC patients and their families, decreases the patient care burden, and reduces the risk of difficult-to-treat infections. Future targeted diagnostic systems may provide a better differentiation of bacterial from viral infections and support the decision against antibiotic therapies [62]. The radiographic diagnosis of LRTIs in this patient population is complicated by the presence of, for example, scoliosis and chronic atelectasis. A CXR can most likely help to rule out an LRTI [63].

### 4.4. Recurrent Pneumonia and Preventive Interventions

Five patients received an inpatient treatment for a bacterial LRTI more than twice during the study period (range: 2–5 admissions). Due to the frequency and severity of the disease in patients with SNIs, practitioners may be inclined to offer a prophylactic therapy to these patients. There is inconclusive evidence that antibiotic prophylaxis in pediatric patients with a high risk of lower respiratory tract infections can reduce the rate of infections, hospital admissions, or mortality [64]. Retrospective data have also demonstrated no beneficial effect of oral secretion management, gastric acid suppression, gastrostomy tube placement, or chest physiotherapy on the recurrence of severe pneumonia in children with a neurological impairment. Professional dental care alone was associated with a lower recurrence of severe lower respiratory tract infections [10]. Although there is evidence of the positive impact of dental hygiene on the incidence of aspiration pneumonia [65], there is a lack of controlled studies to support a clear association [66]. Clinical trials should be conducted to evaluate the efficacy of prophylactic measures in preventing lower respiratory tract infections, particularly in pediatric patients with an SNI.

### 4.5. Disease Burden of Pneumonia in SNI Patients and a Pediatric Palliative Care Perspective

From a PPC perspective, recurrent LRTIs represent a substantial burden and a common cause of death in children and adolescents with life-limiting SNI [28,29,30]. Future evidence-based guidelines and adequate initial therapy could thus reduce acute suffering (e.g., due to dyspnea, prolonged inpatient treatment, or repeated venous punctures) and improve the health-related quality of life for this patient population. For patients with recurrent bacterial LRTIs who are not treatable with oral medications, early implantation of a central venous access device, such as a port-a-cath catheter, is reasonable. In addition, this also facilitates continuing out-of-hospital care for these PPC patients. To prevent severe courses of viral LRTIs, regular immunization (e.g., seasonal influenza, SARS-CoV-2) is also recommended in this patient group [67].

An early adequate treatment could also prevent or delay irreversible long-term sequelae such as increased restrictive pulmonary dysfunctions or the formation of progressive atelectasis with oxygen dependence or even continuous respiratory support (high-flow nasal cannula or non-invasive ventilation). Out of concern for severe respiratory infections, many parents avoid allowing their child to engage in age-appropriate participation such as attending kindergarten or school. This development has been exacerbated during the SARS-CoV-2 pandemic. Future improved treatment options and evidence-based preventive interventions could reduce this parental concern and allow children and adolescents to participate more fully in daily life.

### 4.6. Development of Future Guidelines and Lessons Learned from Our Findings

Regarding antibiotic therapies, future guidelines for the treatment of LRTIs in children and adolescents should also include treatment recommendations for patients with SNIs, even if the recommendations are founded on an expert consensus after an intensive and transparent exchange of experiences and arguments (the Delphi method). Future studies should identify patient-related factors that influence the selection of an antimicrobial therapy. Concerning the probability of a successful therapy and the risk of resistance, guidance should be provided regarding which patients should initially be treated with, for example, broad-spectrum penicillin and which should be given an extended-spectrum treatment with ureidopenicillins or cephalosporins.

Based on the findings of our study, definite treatment recommendations were made for the management of LRTIs in patients with an SNI as part of the implementation of an ABS program at our tertiary care children’s hospital. Due to the possibility of (micro-)aspiration, intravenous ampicillin–sulbactam is our preferred antibiotic in patients with an SNI and a bacterial LRTI; when intravenous access is not feasible, we use amoxicillin–clavulanic acid administered orally or by a gastric tube. If there is no clinical improvement after 72 h, the therapy is escalated to intravenous piperacillin–tazobactam, but the results of the initial respiratory samples have to be considered. In ventilated patients or patients with a tracheostomy, we use piperacillin–tazobactam to provide an effective treatment against Gram-negative bacteria. Additional macrolides are prevented when PCR tests do not yield atypical pathogens such as *Mycoplasma pneumoniae* or *B. pertussis*. The results of the respiratory specimens should be considered in the case of treatment failure. All PPC children with an LRTI should receive an antibiotic stewardship consultation at admission or by day 3 of the treatment.

For patients with recurrent difficult-to-treat LRTIs, the treatment recommendations should be available in the patient record. These recommendations should derive from previous successful antibiotic therapies. In case of more than one effective prior antibiotic therapy, we recommend the substance with the lowest potential to develop an antibiotic resistance.

### 4.7. Limitations

The sample of patients recruited was heterogeneous in terms of age, underlying disease, and medication. However, this reflects the broad spectrum of patients with an SNI treated in our PPC unit. Future studies should investigate the rationale of the treating physicians for selecting the respective antimicrobial drugs. The respiratory specimens were collected from different sections of the respiratory tract and were, therefore, of a heterogeneous quality. Future studies should evaluate the quality of the respiratory specimens from the lower respiratory tract of pediatric patients with an SNI.

## Figures and Tables

**Figure 1 children-09-00852-f001:**
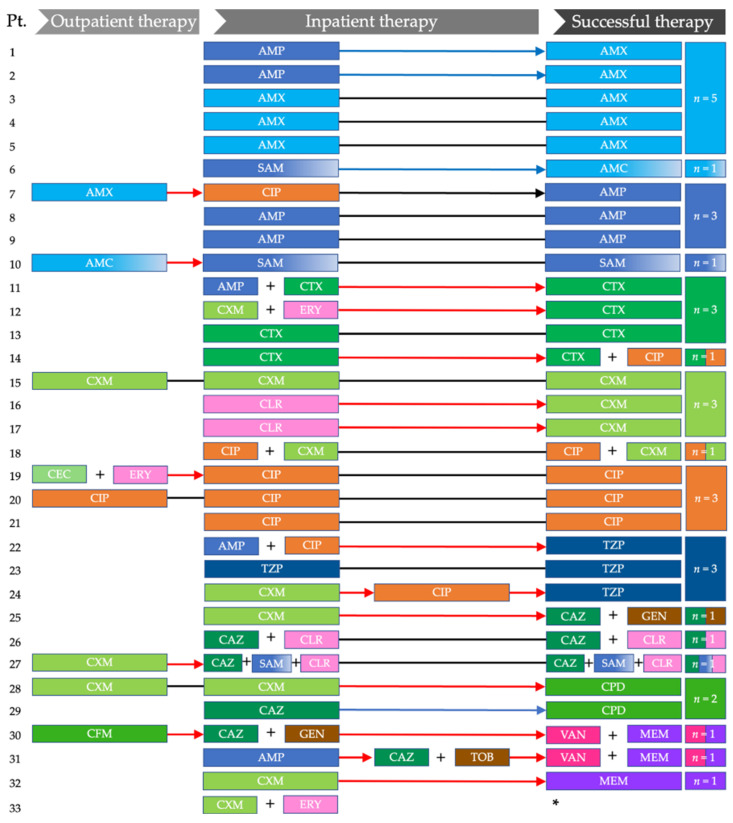
Sequence of antibiotic therapies in *n* = 33 patients ordered by successful therapy. Black lines indicate continuation of the treatment; blue arrows represent sequential intravenous-to-oral antibiotic switch therapy; red arrows indicate a treatment modification. AMX: amoxicillin; AMC: amoxicillin–clavulanic acid; AMP: ampicillin; CAZ: ceftazidime; CEC: cefaclor; CFM: cefixime; CIP: ciprofloxacin; CLR: clarithromycin; CPD: cefpodoxime; CTX: cefotaxime; CXM: cefuroxime; ERY: erythromycin; GEN: gentamicin; MEM: meropenem; Pt: patient; SAM: ampicillin–sulbactam; TOB: tobramycin; TZP: piperacillin–tazobactam; VAN: vancomycin. * Patient died in the course of treatment.

**Table 1 children-09-00852-t001:** Clinical characteristics of the included patients (*n* = 20).

Characteristics	n (%)/Median ± SD or Mean ± SD
**Sex**	
Female	8 (40)
Male	12 (60)
Age, years (median ± SD)	9.6 ± 6.2
Length of hospital stay, days (mean ± SD)	14.5 ± 11.1
**ICD10 2019**	
Q04.9 Congenital malformation of brain, unspecified	8 (40)
E70-E83 Metabolic disorders	2 (10)
G93.1 Anoxic brain damage, not elsewhere classified	3 (15)
P91 Other disturbances of cerebral status of newborn	2 (10)
G40.0 Localization-related idiopathic epilepsy and epileptic syndromes with seizures of localized onset	1 (5)
G71.2 Congenital myopathies	1 (5)
Q87.1 Congenital malformation syndromes predominantly associated with short stature	1 (5)
Q89 Other congenital malformations, not elsewhere classified	1 (5)
Q91 Edwards syndrome and Patau syndrome	1 (5)
**Additional symptoms**	
Epilepsy	18 (90)
Cerebral Palsy	12 (60)
**Death**	2 (21.2)
**MDRO colonization on admission**	3 (7.8)
*Escherichia coli* (rectal)	1 (2.6)
*Morganella morganii* (rectal)	1 (2.6)
MRSA (rectal and respiratory specimen)	1 (2.6)

ICD: International Classification of Diseases, MDRO: multidrug-resistant organisms; MRSA: methicillin-resistant *Staphylococcus aureus*. Continuous variables are shown as median ± SD or mean ± SD and counts as *n* (% of all included patients).

**Table 2 children-09-00852-t002:** Respiratory specimen and isolated bacteria (n = 22).

Respiratory Specimen and Isolated Bacteria	n (%)
Type of sample	
Sputum	9 (22)
Tracheal secretion	8 (19.5)
Oropharyngeal secretion	3 (7.2)
Bronchial secretion	2 (4.9)
**Culture result**	
No bacterial growth	7 (31.8)
**Isolated bacteria (*n* = 23)**	
*Pseudomonas aeruginosa*	3 (13)
*Escherichia coli*	3 (13)
*Klebsiella pneumoniae*	3 (13)
*Haemophilus influenzae*	2 (8.7)
*Proteus mirabilis*	2 (8.7)
*Serratia marcescens*	2 (8.7)
*Achromobacter xylosoxidans*	1 (4.6)
*Acinetobacter baumannii*	1 (4.6)
*Enterobacter aerogenes*	1 (4.6)
*Enterobacter kobei*	1 (4.6)
Group C *Streptococcus*	1 (4.6)
MRSA	1 (4.6)
*Staphylococcus aureus*	1 (4.6)
*Streptococcus* spp.	1 (4.6)

MRSA: methicillin-resistant *Staphylococcus aureus*; spp.: species pluralis. Counts as *n* (% of all specimens/bacteria).

**Table 3 children-09-00852-t003:** Antimicrobial therapies.

Antimicrobial Therapy	n (%)
Outpatient antibiotic therapy *n* = 8	
Cefuroxime	2 (25)
Amoxicillin	1 (12.5)
Amoxicillin-clavulanic acid	1 (12.5)
Cefaclor	1 (12.5)
Cefaclor and erythromycin	1 (12.5)
Cefixime	1 (12.5)
Ciprofloxacin	1 (12.5)
**Initial inpatient antibiotic therapy *n* = 33**	
Ampicillin	5 (15.2)
Cefuroxime	5 (15.2)
Ciprofloxacin	4 (12.1)
Amoxicillin	3 (9.1)
Clarithromycin	2 (6.1)
Ampicillin-sulbactam	2 (6.1)
Cefotaxim	2 (6.1)
Ampicillin and cefotaxim	1 (3)
Amoxicillin and ciprofloxacin	1 (3)
Ceftazidim	1 (3)
Ceftazidim and clarithromycin	1 (3)
Ceftazidim and ampicillin-sulbactam and clarithromycin	1 (3)
Ceftazidim and gentamicin	1 (3)
Cefuroxime and clarithromycin	1 (3)
Cefuroxime and erythromycin	1 (3)
Cefuroxime and ciprofloxacin	1 (3)
Piperacillin/tazobactam	1 (3)
**Successful antibiotic therapy *n* = 32**	
Amoxicillin	5 (15.6)
Ampicillin	3 (9.4)
Cefotaxim	3 (9.4)
Cefuroxime	3 (9.4)
Ciprofloxacin	3 (9.4)
Piperacillin-tazobactam	3 (9.4)
Cefpodoxime	2 (6.3)
Meropenem and vancomycin	2 (6.3)
Amoxicillin-clavulanic acid	1 (3.1)
Ampicillin-sulbactam	1 (3.1)
Cefotaxim and ciprofloxacin	1 (3.1)
Ceftazidim and ampicillin-sulbactam and clarithromycin	1 (3.1)
Ceftazidim and clarithromycin	1 (3.1)
Ceftazidim and gentamicin	1 (3.1)
Ciprofloxacin and cefuroxime	1 (3.1)
Meropenem	1 (3.1)

Counts as *n* (% of included cases).

## Data Availability

The data sets used and analyzed during this study are available from the corresponding author on reasonable request.

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
