# Peer review of "Lower Respiratory Tract Infections in Pediatric Patients with Severe Neurological Impairments: Clinical Observations and Perspectives in a Palliative Care Unit"

_children, 2022, doi:10.3390/children9060852_

Round 1
Reviewer 1 Report
1. line 138 "..12 patients..." add "(%)"
2. line 162 it's "Table S2", should be "Table 2"
3. Table 2 should be reorganized to be more clear
4. line 178 "ciprofloxacin (n=2)" - what this number concerns?"
5. When did you change therapy? Just after hospital admission or later - after culture result receiving? (lines 175-182)
6. line 183 "In n=24 (72.7%) cases..." - reject "n="
7. line 344 "Gram negative..." - should be "Grat negative bacteria"
8. lines 344-345 - Did you write about culture of atypical bacteria? They need special medium so it's difficult to get positive result. Maybe you did other tests to get result about atypical bacteria infection?
Author Response
Dear Reviewer, Thank you for your thoughtful review of our manuscript. We are pleased to have the opportunity to address your concerns. In the following, we address your comments point-by-point. We are optimistic that you will agree that the revisions have improved the manuscript.
- line 138 "..12 patients..." add "(%)"
We have added the information.
- line 162 it's "Table S2", should be "Table 2"
We refer to the supplement Table S1 here, so the designation should be correct.
- Table 2 should be reorganized to be more clear
We have tried to make the table clearer within the journal specifications.
- line 178 "ciprofloxacin (n=2)" - what this number concerns?"
We have removed the number of cases.
- When did you change therapy? Just after hospital admission or later - after culture result receiving? (lines 175-182)
Thank you for pointing this out. We are referring here to therapy changes on admission. We have supplemented the information.
- line 183 "In n=24 (72.7%) cases..." - reject "n="
Thank you for pointing this out. We have changed the text.
- line 344 "Gram negative..." - should be "Grat negative bacteria"
Thank you for pointing this out. We have changed the text line.
- lines 344-345 - Did you write about culture of atypical bacteria? They need special medium so it's difficult to get positive result. Maybe you did other tests to get result about atypical bacteria infection?
Thank you for pointing this out. We referred to PCR diagnostics at this point. We have specified this in more detail now.
Reviewer 2 Report
The authors of this study conducted a retrospective study on the lower respiratory tract infection and response of PPC patinets with SNI, which was of great significance for understanding the disease hazards and management of this group. However, the description in the text was confusing and not very clear, such as the time period of the study (two time periods in the text), the population, and the type of testing. The results of the studies were stacked, making it difficult to clarify. The article need to be better reorganized.
Author Response
The authors of this study conducted a retrospective study on the lower respiratory tract infection and response of PPC patients with SNI, which was of great significance for understanding the disease hazards and management of this group. However, the description in the text was confusing and not very clear, such as the time period of the study (two time periods in the text), the population, and the type of testing. The results of the studies were stacked, making it difficult to clarify. The article needs to be better reorganized.
Dear Reviewer, Thank you for your thoughtful review of our manuscript. We are pleased to have the opportunity to address your concerns.
The study period between February 1st, 2018, and January 31st, 2020 was given once in the Materials and Methods section. In the following, we always refer to this two-year period. We could not find any alternative study time in the manuscript. If there are any remaining uncertainties, please provide us with feedback.
We have added subheadings to the results and discussion section, this should make the results clearer to understand. In addition, we have restructured the discussion of our results to reflect the new subheadings. We are optimistic that you will agree that the revisions have improved the manuscript.
Round 2
Reviewer 2 Report
The article had a certain significance for the study of lower respiratory tract infection in patients with severe neurologic impairment, but there were many problems in the current description of the article that were confusing. The first row of each table was misused.
Author Response
Dear Reviewer, thank you again for your review of our manuscript. We have again streamlined the introduction and methodology section of the manuscript to improve the clarity of the article. Furthermore, we have adjusted the tables, thank you for the comment. Unfortunately, we are currently unable to make any further changes concerning your comments due to a lack of clarification as to what specific changes are to be made. However, we are optimistic that you will agree that the revisions have improved the manuscript.